# Mitigating deep double descent by concatenating inputs

## Abstract

The double descent curve is one of the most intriguing properties of deep neural networks. It contrasts the classical bias-variance curve with the behavior of modern neural networks, occurring where the number of samples nears the number of parameters. In this work, we explore the connection between the double descent phenomena and the number of samples in the deep neural network setting. In particular, we propose a construction which augments the existing dataset by artificially increasing the number of samples. This construction empirically mitigates the double descent curve in this setting. We reproduce existing work on deep double descent, and observe a smooth descent into the overparameterized region for our construction. This occurs both with respect to the model size, and with respect to the number epochs.

## 1 Introduction

Underparameterization and overparameterization are at the heart of understanding modern neural networks. The traditional notion of underparameterization and overparameterization led to the classic U-shaped generalization error curve (Trevor Hastie & Friedman, 2001; Stuart Geman & Doursat, 1992), where generalization would worsen when the model had either too few (underparameterized) or too many parameters (overparameterized). Correspondingly, it was expected that an underparameterized model would underfit and fail to identify more complex and informative patterns, and an overparameterized model would overfit and identify non-informative patterns. This view no longer holds for modern neural networks.

It is widely accepted that neural networks are vastly overparameterized, yet generalize well. There is strong evidence that increasing the number of parameters leads to better generalization (Zagoruyko & Komodakis, 2016; Huang et al., 2017; Larsson et al., 2016), and models are often trained to achieve zero training loss (Salakhutdinov, 2017), while still improving in generalization error, whereas the traditional view would suggest overfitting.

To bridge the gap, Belkin et al. (2018a) proposed the double descent curve, where the underparameterized region follows the U-shaped curve, and the overparameterized region smoothly decreases in generalization error, as the number of parameters increases further. This results in a peak in generalization error, where a fewer number of samples would counter-intuitively decrease the error. There has been extensive experimental evidence of the double descent curve in deep learning (Nakkiran et al., 2019; Yang et al., 2020), as well as in models such as random forests, and one layer neural networks (Belkin et al., 2018a; Ba et al., 2020).

One recurring theme in the definition of overparameterization and underparameterization lies in the number of neural network parameters relative to the number of samples (Belkin et al., 2018a; Nakkiran et al., 2019; Ba et al., 2020; Bibas et al., 2019; Muthukumar et al., 2019; Hastie et al., 2019). On a high level, a greater number of parameters than samples is generally considered overparameterization, and fewer is considered underparameterization.

However, this leads to the question "*What is a sample?*" In this paper, we revisit the fundamental underpinnings of overparameterization and underparameterization, and stress test when it means to be overparameterized or underparameterized, through extensive experiments of a cleverly constructed input. We artificially augment existing datasets by simply stacking every combination of inputs, and show the mitigation of the double descent curve in the deep neural network setting. We

humbly hypothesize that in deep neural networks we can, perhaps, artificially increase the number of samples without increasing the information contained in the dataset, and by implicitly changing the classification pipeline mitigate the double descent curve. In particular, the narrative of our paper obeys the following:

- We propose a simple construction to artificially augment existing datasets of size $O(n)$ by stacking inputs to produce a dataset of size $O(n^2)$.

- We demonstrate that the construction has no impact on the double descent curve in the linear regression case.

- We show experimentally that those results on double descent curve do not extend to the case of neural networks.

Concretely, we reproduce results from recent landmark papers, and present the difference in behavior with respect to the double descent curve.

## 2 RELATED WORKS

The double descent curve was proposed recently in (Belkin et al., 2018a), where the authors define overparameterization and underparameterization as the proportion of parameters to samples. The authors explain the phenomenon through the model capacity class. With more parameters in the overparameterized region, there is larger "capacity" (i.e., the model class contains more candidates), and thus may contain better, simpler models by Occam's Razor rule. The interpolation region is suggested to exist when the model capacity is capable of fitting the data nearly perfectly by overfitting on non-informative features, resulting in higher test error. Experiments included a one layer neural network, random forests, and others.

The double descent curve is also observed in deep neural networks (Nakkiran et al., 2019), with the additional observation of epoch-wise double descent. Experimentation is amplified by label noise. With the observation of unimodel variance (Neal et al., 2018), Yang et al. (2020) also decomposes the risk into bias and variance, and posits that the double descent curve arises due to the bell-shaped variance curve rising faster than the bias decreases.

There is substantial theoretical work on double descent, particularly in the least squares regression setting. Advani & Saxe (2017) analyses this linear setting and proves the existence of the interpolation region, where the number of parameters equals the number of samples in the asymptotic limit where samples and parameters tend to infinity. Hastie et al. (2019) follows a similar line of work, and proves that regularization reduces the peak in the interpolation region. Belkin et al. (2019b) requires only finite samples, where the features and target be jointly Gaussian. Other papers with similar setup include (Bartlett et al., 2019; Muthukumar et al., 2019; Bibas et al., 2019; Mitra, 2019; Mei & Montanari, 2019).

Ba et al. (2020) analyses the least squares regression setting for two layer linear neural networks in the asymptotic setting, where the double descent curve is present when only the second layer is optimized. There is also work in proving that optimally tuned $\ell_2$-norm regularization mitigates the double descent curve for certain linear regression models with isotropic data distribution (Nakkiran, 2019). This setting has also been studied with respect to the variance in the parameter space (Bartlett et al., 2019). Multiple descent has also been studied, and in particular there is work to show in the linear regression setting that multiple descent curves can be directly designed by the user (Chen et al., 2020). Additionally, there is supporting evidence of double descent in the sample-wise perspective (Nakkiran et al., 2020).

There is other work in this area, including studying the double descent curve for least squares in random feature models (Belkin et al., 2019a; d'Ascoli et al., 2020; Ghorbani et al., 2019), leveraging the Neural Tangent Kernel to argue for certain number of parameters the output of the neural network diverges (Geiger et al., 2020), characterizing the double descent in non-linear settings (Caron & Chretien, 2020), kernel learning (Belkin et al., 2018b; Liang et al., 2019), and connecting to other fields (Geiger et al., 2019). Lastly, we note here that, in the deep neural network setting, models can be trained to zero training loss even with random labels (Zhang et al., 2016).

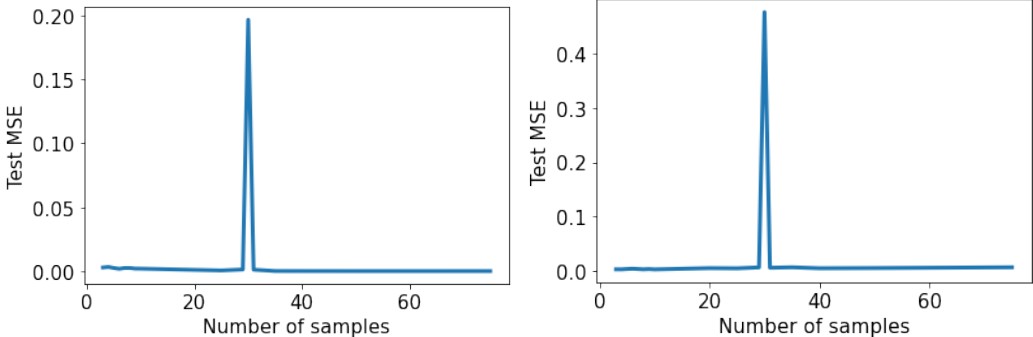

Figure 1: Left: The standard case. Right: The concatenated inputs construction. Plots of the Test MSE verses number of samples (pre-concatenation) for min-norm ridgeless regression, where $d = 30$. Following Nakkiran (2019), inputs are drawn $x \sim \mathcal{N}(0, I_d)$, target $y = \theta x + \mathcal{N}(0, \sigma^2)$, where $\theta$ are the parameters, $||\theta||_2 = 1$, $\sigma = 0.1$. $\hat{\theta} = X^\dagger y$.

## 3 THE CONCATENATED INPUTS CONSTRUCTION

We introduce the *concatenated inputs construction*, on which our main hypothesis is based on. The concatenated inputs construction refers to the general idea of concatenating pairs of inputs and element-wise adding and averaging pairs of outputs to produce new inputs and targets. This way the size of a dataset can be artificially –but non-trivially– increased.

This construction can be applied both to the regression setting and the classification setting. In the setting of linear regression, for given input pairs, $(x_1, y_1)$, $(x_2, y_2)$, an augmented dataset can be constructed as:

$$\left\{ \left([x_1, x_1], \tfrac{y_1+y_1}{2}\right), \left([x_1, x_2], \tfrac{y_1+y_2}{2}\right), \left([x_2, x_1], \tfrac{y_2+y_1}{2}\right), \left([x_2, x_2], \tfrac{y_2+y_2}{2}\right) \right\},$$

where $[\alpha, \beta]$ represents concatenation of the input $\alpha, \beta$. In the setting of classification, the process is identical, where the targets are produced by element-wise addition and then averaged to sum to 1. The averaging is not strictly necessary even in the deep neural network classification case, where the binary cross entropy loss can be used instead of cross entropy. For test data, we concatenate the same input with itself, and the target is the original target. This way a dataset of size $O(n)$ is artificially augmented to size $O(n^2)$.

Concretely, our reasons for the concatenated inputs construction are as follows: $i)$ there is limited injection of information or semantic meaning; $ii)$ the number of samples is significantly increased. For the purposes of understanding underparameterization, overparameterization and the double descent curve, such a construction tries to isolate the definition of a sample. We revisit and assess these implications in the context of extensive experiments in the following sections.

## 4 RESULTS

In this section, we reproduce settings from benchmark double descent papers, add the concatenated inputs construction and analyze the findings. In particular, we begin with the linear regression, move to one hidden layer feedforward neural networks, and then deep neural networks, in both the model parameter-wise double descent and epoch-wise double descent. Finally, we analyze the performance of deep neural networks for the concatenated inputs construction, and the behavior of the double descent curve in the classification setting, when we transfer from the cross entropy to the binary cross entropy loss.

### 4.1 LINEAR REGRESSION

The linear regression setting has been a fruitful testbed for empirical work in double descent, as well as yielding substantial theoretical understanding. The concatenated inputs construction is applied

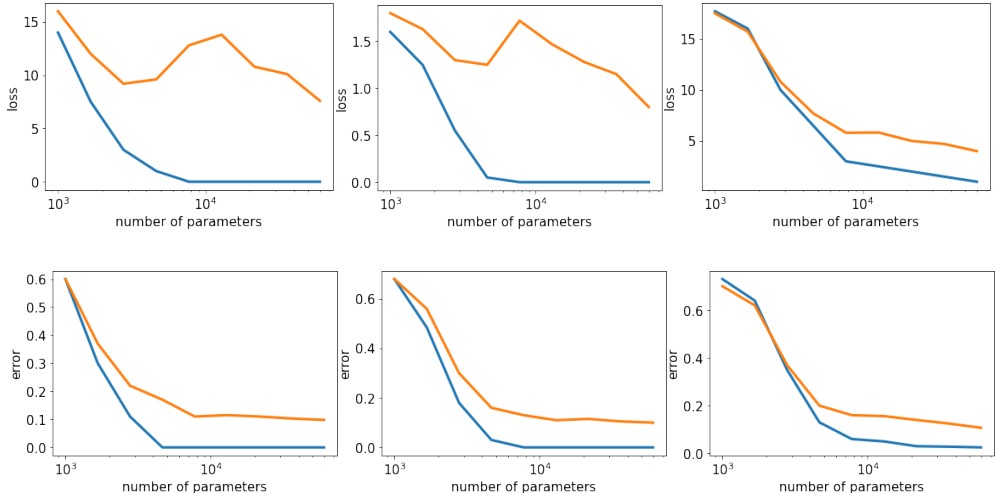

Figure 2: Top row: Loss against number of parameters. Bottom row: Error against number of parameters. In order from left to right: 1. Input is $32 \times 32$ image, label is one-hot vector and loss is Cross Entropy, 2. Input is $64 \times 32$ image (two of the same image stacked on top of each other), label is one-hot vector and loss is Cross Entropy, 3. Input is $64 \times 32$ image (two different $32 \times 32$ images stacked on top of each other), label is two-hot vector of values 0.5 and 0.5 and loss is Cross Entropy. All models trained with batch size of 100 for 1000 epochs on a subset ($n = 4000$) of the MNIST dataset. Adam optimizer is used with learning rate = 0.001, $\beta_1 = 0.9$, $\beta_2 = 0.999$. Models are feedforward neural networks with a single layer of hidden units with ReLU activation.

similarly here, however with different motivation. Namely, we wish to motivate that the concatenated inputs construction is not expected to add any information and is therefore not expected to impact the double descent curve.

We reproduce the linear regression setting from Nakkiran (2019), given in Figure 1. For the concatenated inputs construction, we first draw the number of samples before concatenation and construction of the augmented dataset. We observe that, by construction, the concatenated inputs construction does not affect the double descent curve, and the peak occurs in the exact same location. We also make the remark here that it is not surprising that this is the case, and it is not complicated to understand why from a theoretical perspective.

### 4.2 ONE HIDDEN LAYER FEEDFORWARD NEURAL NETWORK

Following linear regression, we move to neural networks. We train a feedforward neural network with one hidden layer and ReLU activations on a subset of the MNIST dataset, reproducing the experimental setup from Belkin et al. (2018a). We vary the number of parameters in the neural network by changing the size of the hidden layer. We use the Cross Entropy loss instead of the original MSE loss due to the prevalence of those losses for image classification tasks. This is shown in Figure 2.

We observe the double descent in the loss versus number of parameters, but were unable to produce the double descent in the error. In the rightmost plot, the double descent curve is completely removed in the concatenated inputs construction relative to the other two settings. Namely, a smooth decrease in loss is observed, where there is a clear double descent in the other cases. Furthermore, we provide the extra setting of concatenating each input only by itself, and the double descent curve is present almost exactly in this scenario. *This provides further evidence that the disappearance of the double descent is not due to the extra parameters which originate from the larger sized inputs.* In this setting, it appears that the behavior of underparameterization and overparameterization can be altered by simply artificially increasing the number of samples through concatenating images.

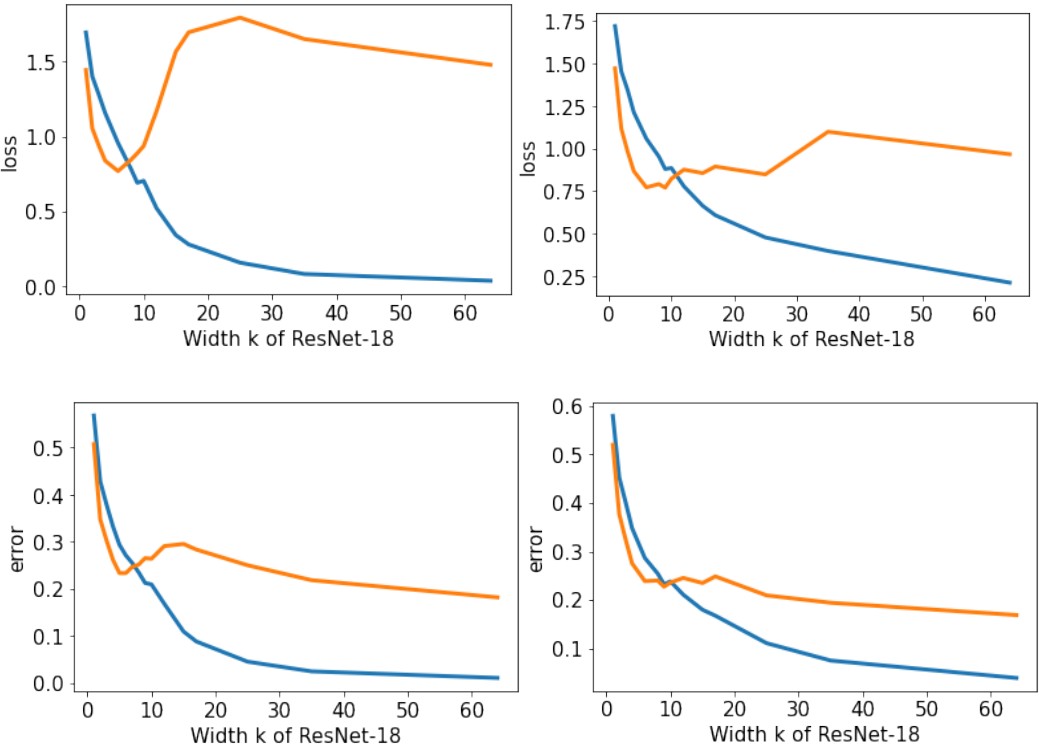

Figure 3: Left: Standard one-hot vector setup, Right: Concatenated inputs construction. Top: Loss. Bottom: Error. All models trained with batch size of 128 for 400 epochs with 15% label noise on the CIFAR10 dataset. Label noise is defined by changing p% of the samples to random wrong labels and is applied prior to training, and prior to the concatenated inputs construction. Adam optimizer is used with learning rate = 0.0001, $\beta_1 = 0.9$, $\beta_2 = 0.999$. Models are ResNet18 architecture where the width $k$ is varied. $k = 64$ is the standard ResNet18. The plot is visually similar when plotted against parameters, since concatenated inputs construction typically adds less than 5% of the total parameters.

In addition, the model trained on MNIST and one-hot vectors can be concatenated with itself, with all other parameters being zero, to produce a model with two times the number of hidden units which can be applied to the concatenated inputs construction. We consider this setting in the context of a possible explanation of the interpolation region, where the number of parameters nears that of samples. Concretely, it is possible for a neural network with double the hidden units in the concatenated inputs construction to recover the double descent curve by learning two smaller, disconnected networks, where each of the smaller networks are the ones learned in the double descent peak of the standard, one-hot case. However, in practice while the network can do so, it does not appear to, which leads to the smooth descent in the rightmost plot in Figure 2.

## 4.3 DEEP NEURAL NETWORKS

In this section, we investigate deep double descent in several settings, including model size double descent, and epoch-wise double descent. Finally, we decompose the loss into bias and variance to further investigate the effect of the concatenated inputs construction.

### 4.3.1 MODEL SIZE DOUBLE DESCENT

**ResNet18 - CIFAR10.** We train a ResNet18 architecture on the CIFAR10 dataset in Figure 3, reproducing the experimental setup from Nakkiran et al. (2019). We vary the number of parameters in the neural network by changing the width $k$. In this experimental setup, we use deep neural networks and reproduce the double descent with respect to the model size. Immediately, we see that the double descent curve is relatively mitigated in the case of the concatenated inputs construction

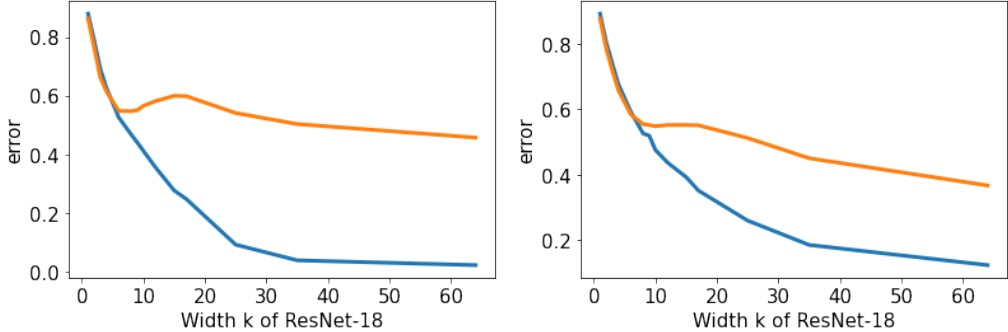

Figure 4: Left: Standard one-hot vector setup, Right: Concatenated inputs construction. All models trained with batch size of 128 for 400 epochs with 15% label noise on the CIFAR100 dataset. Label noise is defined by changing p% of the samples to random wrong labels and is applied prior to training, and prior to stacking for two-hot vectors. Adam optimizer is used with learning rate = 0.0001, $\beta_1 = 0.9$, $\beta_2 = 0.999$. Models are ResNet18 architecture where the width $k$ is varied.

where from $k = 5$ to 20. Here, the curve is much smoother, although not entirely smooth. Notably, the concatenated inputs construction retains the test error across $k$, except in the interpolation region where it achieves significantly lower test error ($> 5\%$). We also note that the conclusion is identical when error is plotted against parameters, since the concatenated inputs construction typically adds a very small number of parameters ($< 5\%$).

We also present the plot for loss in Figure 3 (Top row). Here, we observe a clear double descent in the one-hot case, where the model follows the classical U shaped curve prior to the interpolation region, and then steadily decreases in loss after. The concatenated inputs construction exhibits more interesting behavior where the U curve is significantly muted and mitigated, and there is no clear peak of the curve.

**ResNet18 - CIFAR100.** We train a ResNet18 architecture on the CIFAR100 dataset in Figure 4, reproducing the experimental setup from Nakkiran et al. (2019) and almost identical setup in `ResNet18 - CIFAR10`. The results are similar and slightly clearer here, where the double descent curve in error entirely disappears with the concatenated inputs construction, and test error is decreased in the interpolation region. Again, the test error difference is significant ($> 5\%$).

### 4.3.2 EPOCH-WISE DOUBLE DESCENT

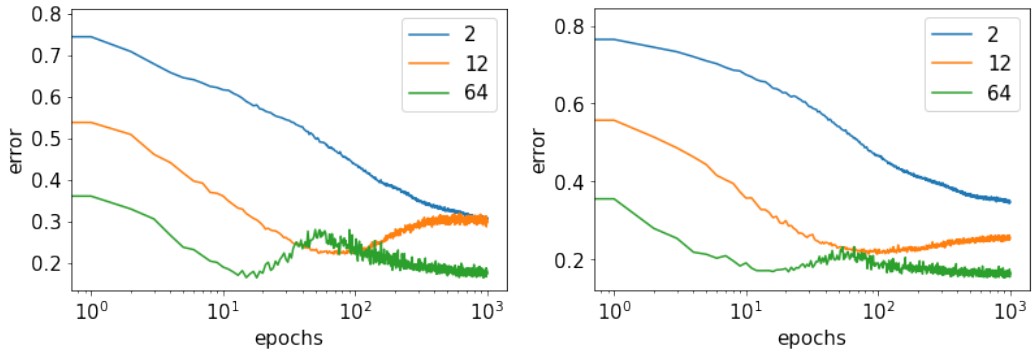

Figure 5: Left: Standard one-hot vector setup, Right: Concatenated inputs construction. Models are ResNet18 architecture on CIFAR-10 where the error during training is plotted above.

**ResNet18 - CIFAR10.** We use the same setting as in the previous section, except we train models for an additional 600 epochs, for a total of 1000 epochs. In Figure 5, we plot test error against epochs. In the one hot setting, as expected we observe a U shape for medium sized models and a

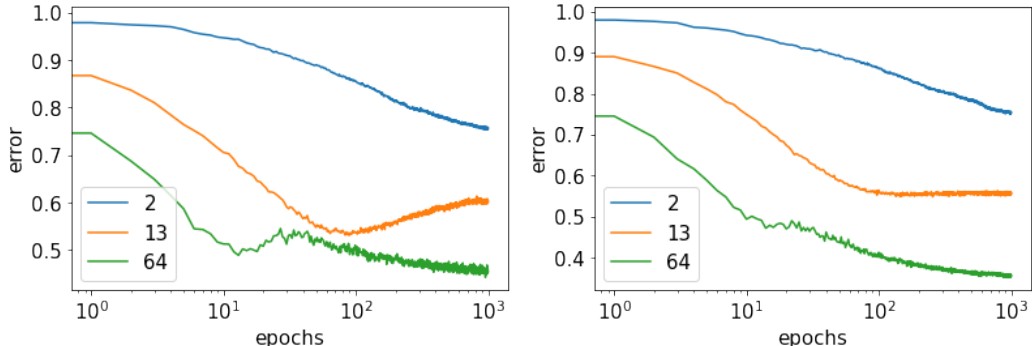

Figure 6: Left: Standard one-hot vector setup, Right: Concatenated inputs construction. Models are ResNet18 architecture on CIFAR-100 where the error during training is plotted above.

double descent for larger models (Nakkiran et al., 2019). For the concatenated inputs construction, a flat U shape and double descent is indeed observed, although they are significantly mitigated. The mitigation allows a 5% improvement at the end of training for medium sized models and a 10% improvement in the middle of training for large sized models.

**ResNet18 - CIFAR100.** Similarly to above, we use the same setting as in the previous section, except we train models for a total of 1000 epochs. Results are given in Figure 6. The one hot results mirror ResNet18 - CIFAR10, as expected (Nakkiran et al., 2019). For the concatenated input construction, we observe complete mitigation of the double descent curve for medium and large sized models. For large models, this is more than a 10% gap.

### 4.3.3 BIAS VARIANCE DECOMPOSITION

In this section, we follow Yang et al. (2020) and decompose the loss into bias and variance. Namely, let CE denote the cross entropy loss, $T$ a random variable representing the training set, $\pi$ is the true one-hot label, $\bar{\pi}$ is the average log-probability after normalization, and $\hat{\pi}$ is the output of the neural network. Then,

$$\mathbb{E}_T[\text{CE}(\pi, \hat{\pi})] = D_{KL}(\pi||\bar{\pi}) + \mathbb{E}_T[D_{KL}(\bar{\pi}||\hat{\pi})]$$

where the first component is the bias and the second component is the variance. On a high level, the variance can then be estimated by training separate same capacity models on different splits of the dataset, and then measuring the difference in outputs on the same test set. The bias is then computed by subtracting the empirical variance from the empirical risk. For finer details, see Yang et al. (2020)

For training, we follow Yang et al. (2020) and train a ResNet-34 (He et al., 2016) on the CIFAR10 dataset, with stochastic gradient descent (learning rate = 0.1, momentum = 0.9). The learning rate is decayed a factor of 0.1 every 200 epochs, and a weight decay of 5e-4 is used. The width $k$ of the network is varied suitably between 1 and 64. We also make 5 splits of 10,000 training samples for the calculation of bias and variance.

We present results in Figure 7. The concatenated inputs construction significantly delays and smoothens the increase in variance relative to the standard case, where the unimodal variance is significantly sharper. This impacts the shape of the test error, where in this setting we see a shifted bump in test error for the concatenated inputs construction. One possible explanation is in the case of deep neural networks the concatenated inputs construction is a form of implicit regularization for small models, which controls overfitting and leads to a smoother variance curve.

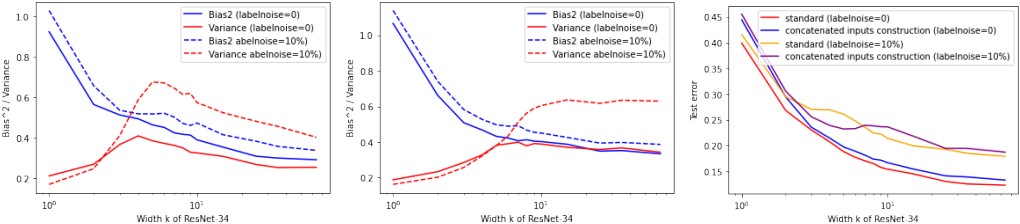

Figure 7: Left: Bias and Variance against width k of ResNet-34 in the standard case. Middle: Bias and Variance against width k of ResNet-34 in the concatenated inputs construction. Right: Test error against width k of ResNet-34 in the standard case and concatenated inputs construction.

## 5 DISCUSSION

We revisit the topic of overparameterization, underparameterization and the double descent curve. The understanding of overparameterization, underparameterization, and the double descent curve is strongly tied to the number of samples. It appears that given a fixed number of unique samples it is possible to manipulate overparameterization and underparameterization by artificially boosting the number of samples by the concatenated inputs construction. This is based on the observation of removing or shifting the double descent curve through the concatenated inputs construction.

A similar topic on double descent is the possible explanation that the model is being forced to fit the training data as perfectly as possible, and at some model capacity it is possible to fit the training data perfectly by overfitting on non-existent, or weakly present, features. This results in overfitting and the double descent curve. Interestingly, the concatenated inputs construction generally removes the double descent curve, even though it is possible to build models for the concatenated inputs construction from models for the standard setting. This suggests a possible route to improve on the understanding of the relationship to the model capacity.

Furthermore, we consider the topic of underfitting. It is well know that overparameterized neural networks don't exhibit strong overfitting in practice, even though they can memorize the dataset (Zhang et al., 2016). The experimental results in this work regarding underparameterization, overparameterization and the double descent curve support that the behavior of the neural network can change with respect to the number of samples, even if the majority of samples add limited information, via the concatenated inputs construction. In this view, the concatenated inputs construction creates possibly a huge dataset, for example $50,000^2$ samples for the originally $50,000$ samples CIFAR10 dataset where $50,000^2 = 2,500,000,000$ is far larger than any neural network for the CIFAR10 dataset. Yet, there is no noticeable underfitting. Namely, the concatenated inputs construction very quickly achieves comparable performance compared to the standard one-hot vector deep learning setting. This suggests we may need to rethink the relationship between underfitting and the number of parameters, samples, and model capacity.

## 6 CONCLUSION

In this paper, we examine the double descent phenomena through the concatenated inputs construction. The concatenated inputs construction is designed to add limited information whilst artificially augmenting the size of the dataset. As constructed, in the linear regression setting there is no impact on the location of the peak. Yet, in the neural network setting, the concatenated inputs construction generally removes the double descent curve. Finally, we explore this phenomena through bias variance decomposition, and draw connections with samples, model capacity, and underfitting in discussion.

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
