# OpenReview forum: "Mitigating Deep Double Descent by Concatenating Inputs"
_ICLR.cc/2021/Conference — Reject_

### Official Review · AnonReviewer3 · 2020-10-20
**Interesting idea but the contribution appears not solid enough**

**Rating:** 4
**Confidence:** 3

**Review:**

**Summary**: In this article, the authors proposed a data augmentation procedure by concatenating the input data to produce an augmented dataset of size $O(n^2)$ from an original dataset of size $O(n)$, so as to mitigate the double descent curve. The authors showed experimentally that such construction does not impact the double descent curves in linear regression but can largely alleviate those in the case of (deep) neural nets.

**Strong points**: The idea of (artificial) data augmentation looks interesting. Experiments are conducted on linear regression, single-hidden-layer ReLU network, as well as on ResNet18.

**Weak points**: The organization and presentation of the article can be significantly improved, there are many places in the paper where things are not clearly explained and properly compared to the existing literature. Both the theoretical and empirical contributions of this work are somewhat limited. See **detailed comments** below.

**Recommendation**: The idea in this paper looks interesting. However, there is visually no theoretical contribution and the empirical contribution is also somewhat limited. In general, this paper does not meet the standards to be published at ICLR.

**Detailed comments**:

* P3 Sec 3: just some comments on the proposed "concatenated inputs construction": (i) at the same time of augmenting the size of the dataset, this procedure also increases the **dimensionality** of the data (by a factor of 2 I think); and (ii) this procedure implicitly introduces, at least in the case of regression, some sort of invariance or implicit bias in the dataset. For instance, here the input $[x_1, x_2]$ and $[x_2, x_1]$ correspond to **exactly** the same target $(y_1 + y_2)/2$. And I think this imposes some structural constraints in the trained network which may help "mitigate" the double descent test curve?
* Sec 4.1 and Figure 1: "We also make the remark here that it is not surprising that this is the case, and it is not complicated to understand why from a theoretical perspective.": it would be helpful to compare the empirical results here (in the left plot of Figure 1) to the theoretical prediction in for example (Bartlett et al. 2019) or (Hastie et al. 2019), and perhaps with different values of $\sigma^2$: it is somehow less interesting to see an almost straight test MSE line and we clearly do not see the classical U-shaped curve here. Also, I am not sure that the theoretical understanding of the right plot of Figure 1 easily follows from existing results since, the concatenated data are no longer standard Gaussian, and I think they are even not linear transformation or simple Kronecker product of the original Gaussian data.
* Is Figure 2 trying to reproduce Figure 4 in (Belkin et al. 2018a)? Note that the way of counting the number of parameters is slightly different there and I think this explains that under the same setting (training set size $n = 4 \cdot 10^3$) the test loss peaks are observed at different numbers of parameters (around $10^4$ here and around **same** $4 \cdot 10^3$ in (Belkin et al. 2018a)). And I believe the curves could be made more smooth and with confidence intervals.
* Figure 2 top right plot "the double descent curve is completely removed": it would of interest to check whether there is a double-descent-type peak around $O(n^2)$, to help better understand the proposed construction.
* Sec 4.2: it seems to me that the remark "double descent in the loss versus number of parameters, but were unable to produce the double descent in the error" was already mentioned in (Nakkiran et al. 2019).
* Figure 3: even larger width $k$ is expected here for concatenated inputs since the training loss is still at a relatively high level. The same applies to Figure 4.
* The legend, x, and y-axes are hardly visible in Figure 7.

---

### Official Review · AnonReviewer4 · 2020-10-27
**Review of Paper 2427**

**Rating:** 2
**Confidence:** 4

**Review:**

There has been a recent surge of research interest in explaining the double descent phenomenon, which contradicts the classical bias-variance tradeoff and may contribute to the success of DNNs and other overparametrized models. This paper proposes to mitigate double descent by artificially augmenting the dataset with concatenated inputs, and presents some empirical results for linear regression and neural networks.

Although the paper is on a timely and interesting topic, the idea of using data augmentation to mitigate overfitting is certainly not new and has been understood in many areas of machine learning. In fact, augmenting the dataset with duplicates or artificially constructed data is equivalent to incorporating certain prior structure or imposing some form of regularization, which in turn should have an impact on the double descent curve. On the theoretical side, the paper does not provide any rigorous theoretical analysis; on the methodological side, the data augmentation strategy is defined in a restrictive manner and seems to be of little practical value.

More comments:
1) Figures 2-4: It is difficult to decipher these figures without legends. I can’t find anywhere in the text or captions mentioning what the orange and blue curves mean. Also, how are the loss and error defined?

2) The paper argues that the sample size, and hence overparametrization and underparametrization, in the double descent phenomenon is ambiguous and proposes to increase the sample size by artificially augmenting the dataset. In fact, the notion of sample size usually comes with the i.i.d. assumption, i.e., the samples are drawn independently from a common distribution. In this sense, augmenting the dataset with concatenated samples does not directly increase the sample size. Different ways of constructing the concatenated samples lead to different effective sample sizes, which should be properly defined in order to study their impact on the double descent curve.

3) The paper aims to mitigate the double descent curve, which, however, is not necessarily harmful. A more critical question is how the lowest test error in the overparametrized regime compares to the optimal test error in the traditional underparametrized regime. If double descent helps to generalize well, why should we mitigate it?

---

### Official Review · AnonReviewer1 · 2020-10-28
**No convincing reasoning**

**Rating:** 3
**Confidence:** 3

**Review:**

The paper investigates the double descent phenomenon. It proposes the augmentation of the dataset via concatenating the covariate x and interpolating the label y, which increases the data size from n to n^2. The paper shows that the phenomenon of double descent can be mitigated via augmenting the input. The idea of investigating double descent from manipulating samples is novel and interesting.

However, the paper is quite poorly-written. For example, I can't tell what the yellow and blue lines in Figure 2 and later figures represent, what is the difference between loss and error (which is never defined in the paper), and how the figures are generated from the concatenated data. The whole experimental setup is unclear to me. The authors are suggested to provide a concrete introduction and experimental details for it.

Furthermore, the paper does not provide enough reasoning how simply concatenating the x and averaging y can mitigate the double descent phenomenon. First of all, are we supposed to train on the augmented data instead of the original data? Assume that we are running on completely synthetic data where y has a non-linear dependence with x (e.g. in the synthetic data y is the output of 3 layer neural network), then it seems that the pair ([x1,x2], (y1+y2)/2) will only hurt the performance since it adds extra complexity (one more linear transformation on the data) to really fit the model. It is also surprising why before and after concatenation, the error can be that much different with the same network width in Figure 4. The author mentioned at the end of Section 4 that one reason could be that the variance is smoothed due to implicit regularization from concatenation. Is it the case for all the experiments conducted above?

The authors are suggested to analyze the effect of concatenation on double descent thoroughly (e.g. analyze the bias and variance separately in the linear regression case to show why the double descent phenomenon is mitigated), provide convincing reasoning and more experiment details.

---

### Official Review · AnonReviewer2 · 2020-10-28
**The Concatenated input construction is cool but the paper falls short from giving an interesting explanation of the results.**

**Rating:** 5
**Confidence:** 4

**Review:**

Summary:
The paper empirically studies the Double Descent phenomenon by using a cool construction that squares the dataset size by concatenation of every pair of inputs and linear interpolation of the labels. The main empirical finding is that this construction mitigates double descent.

++++++

Main Comments:
The concatenated input construction is a really cool experiment and I enjoyed reading about it. Nevertheless, the empirical findings are not strong enough to warrant publication in my opinion. While I believe that input/label concatenation/interpolation mitigates double descent, this statement by itself is not very interesting to the community. I believe that as the authors note, the concatenated inputs provide some sort of regularization (my intuition that something close to label smoothing/mixmatch is happening) and we already know that regularization mitigates double descent. Moreover, if we look at the paper from the perspective of EMC (Nakkiran 19') the results are not surprising as we not only change the number of samples but also the difficulty of the distribution (and add some parameters to the model).

Pros:
The concatenated input construction is an interesting and original experiment and using it to mitigate double descent is somewhat surprising.

Cons:
While the experiment is original, as of now it is unclear what is causing the mitigation of double descent and what is the broad point the authors try to convey.

Points for improvements:
While mitigating double descent by the concatenated input construction is not a fundamental question by itself, the question that the authors sought to study 'what is a sample' is a very interesting one. I believe that going further down that road can make an interesting and fundamental investigation.

Argument for my assessment:
In my opinion the paper with a better story (and further experimental/theoretical evidence that support it) could be publishable but it falls short at its current state.

Minor Comments:
I would put the MNIST experiments in the appendix. Double Descent is not about test loss which often behaves very differently than test error (cf. figure 2 in https://arxiv.org/abs/1710.10345) and having a convincing experiment on CIFAR is enough.

---

### Decision · Program_Chairs · 2021-01-07
**Final Decision**

**Decision:**

Reject

**Comment:**

This paper proposed an augmentation construction to mitigate the double descent. For any pairs of data points, the constructed input is simply concatenation of two inputs and the constructed label is the average of their corresponding labels. The authors further empirically show that this would mitigate double descent.

Reviewers unanimously like the main idea of the paper but they have other major concerns about this work. The main concern is that we already know double descent is not a practical issue since it can be mitigated by early stopping or proper regularization (Nakkiran 20'). Therefore, the main benefit from this paper could come from a better understanding of double descent using the observations from this construction. However, the paper does not provide us with insightful theoretical or empirical findings beyond the main observation. There are a couple of other concerns as well about discussions around #samples and the fact that the proposed construction is not i.i.d. and also lack of proper discussion about the relationship between the proposed construction and regularization techniques.

Unfortunately, authors did not responded to reviewers concerns. Nonetheless, I encourage authors to read reviewers' specific feedbacks, incorporate them and resubmit their work.

Given the above concerns, I recommend rejecting the paper.